# Genome-Wide Identification of the *BnaRFS* Gene Family and Functional Characterization of *BnaRFS6* in *Brassica napus*

**DOI:** 10.3390/genes16091032

**Published:** 2025-08-29

**Authors:** Bingqian Zhou, Chunyun Guan, Mei Guan

**Affiliations:** 1College of Agriculture, Hunan Agriculture University, Changsha 410128, China; zhoubq202212@stu.hunau.edu.cn (B.Z.); guancy2011@aliyun.com (C.G.); 2Hunan Branch of National Oilseed Crops Improvement Center, Changsha 410128, China; 3 Collaborative Innovation Center of Grain and Oil Crops in South China, Changsha 410128, China

**Keywords:** *Brassica napus*, soluble sugars, raffinose synthase, *BnaRFS6*

## Abstract

Background: Raffinose synthase (RFS) plays a crucial role in plant growth and development, as well as in responses to biotic and abiotic stresses. However, its functions in *Brassica napus* remain poorly understood. Methods: To investigate the characteristics of the *RFS* gene family in *B. napus* (rapeseed), five *Arabidopsis thaliana RFS* gene sequences were used as references to identify thirteen *RFS* genes in *B. napus*, four in *Brassica rapa*, and six in *Brassica oleracea*. A comprehensive analysis was conducted, including molecular characteristics, phylogenetic relationships, conserved protein motifs, gene structures, and chromosomal localization. Results: *BnaC02G0100500ZS* was selected as a candidate gene due to its unique expression profile. Sequence alignment identified it as *BnaRFS6*, and subcellular localization revealed that its encoded protein is localized in the mitochondria. Overexpression of *BnaRFS6* in rapeseed significantly affected the soluble sugar and starch content in the stalks, resulting in increased levels of fructose, glucose, and raffinose, and a decreased starch content. Conclusions: These findings highlight the role of *BnaRFS6* in enhancing sugar metabolism in *B. napus*, particularly in relation to fructose, glucose, and raffinose accumulation. Understanding its potential function provides a foundation for improving the sugar content and taste of rapeseed stalks through genetic engineering in the future.

## 1. Introduction

Rapeseed (*B. napus* L., AACC, 2n = 38) is one of the major oilseed crops in China. Under the current circumstances of declining rapeseed prices, multifunctional utilization has become a key direction and hotspot for the development of the rapeseed industry. It is also an important approach for adjusting the agricultural industrial structure and improving agricultural productivity [1]. Among the various strategies, the development of oilseed-vegetable-dual-purpose (OVDP) varieties is progressing rapidly. The flowering stalks of rapeseed are nutritious and palatable vegetables that are rich in vitamins such as vitamin C, carotenoids, and folate [2]. However, pungent and bitter-tasting compounds such as isothiocyanates—degradation products of glucosinolates [3,4]—and goitrogens [5,6] limit consumer acceptance and consumption of Brassica vegetables. Therefore, increasing the content of soluble sugars and breeding sweet and palatable vegetable-type rapeseed varieties are of great significance.

The accumulation and metabolism of carbohydrates in plants are regulated by a series of key enzymes. As the primary products of photosynthesis, carbohydrates—especially sucrose—are transported via the phloem from source organs (leaves) to sink organs (storage tissues) to promote growth and nutrient absorption in those tissues [7]. In sink organs, sucrose is transformed into different storage forms under the action of multiple metabolic enzymes [8]. The major enzymes involved in this process include sucrose phosphate synthase (SPS), sucrose synthase (SUS), invertase (Inv), fructokinase (FK), hexokinase (HK), and raffinose synthase (RFS). The coordinated activity of these enzymes determines the sugar accumulation patterns and metabolic fluxes in various plant organs, thereby affecting plant growth, development, and biomass accumulation.

Raffinose synthase (RFS, EC 2.4.1.82), a key member of the raffinose family oligosaccharides (RFOs), catalyzes the α-1,6-galactosyl extension of sucrose and serves as a rate-limiting enzyme in the biosynthesis of RFOs [9]. Based on protein sequence homology and phylogenetic analyses, the *RFS* gene family shows considerable diversity across different plant species. In *Arabidopsis thaliana*, six *RFS* genes have been identified—*AtRS1*, *AtRS2*, *AtRS3*, *AtRS4*, *AtRS5*, and *AtRS6*—of which *AtRS3* is a pseudogene [10]. In the *Oryza sativa* genome, six *RFS* genes have also been identified: *OsRS1*, *OsRS2*, *OsRS3*, *OsRS4*, *OsRS5*, and *OsRS6* [11]. In *Zea mays*, four *RFS* genes—*ZmRS1*, *ZmRS2*, *ZmRS3*, and *ZmRS10*—have been reported [12], with *ZmRAFS* being unique to maize and playing a crucial role in seed vigor and longevity [13]. In Arabidopsis, overexpression of *RFS* genes such as *AtRS5* significantly enhances raffinose accumulation, thereby improving drought tolerance [14]. Raffinose and its precursor galactinol tend to accumulate in leaves during abiotic stress conditions. RFS catalyzes the formation of raffinose by transferring a galactosyl group from galactinol to sucrose. In maize, *ZmRAFS* enhances drought tolerance by either synthesizing raffinose or hydrolyzing galactinol, depending on sucrose availability in plant cells [15].

Although the *RFS* gene family has been extensively studied in many plant species, little is known about its roles in *B. napus*. In this study, a comprehensive analysis of the *BnaRFS* gene family was conducted. Based on phylogenetic analysis of protein sequences from *A. thaliana*, *B. rapa*, *B. oleracea*, and *B. napus*, the RFS proteins were classified into three distinct subgroups. Further analyses were performed on the gene structures, chromosomal localization, and conserved motifs of the *RFS* genes. An *RFS* gene exhibiting high expression during the bolting stage was cloned and, based on its sequence homology with Arabidopsis RFS proteins, was designated as *BnaRFS6*. The subcellular localization of the BnaRFS6-encoded protein was examined, and the impact of its overexpression on soluble sugar content in the rapeseed stalk was investigated.

## 2. Results

### 2.1. Bioinformatic Analysis of the RFS Gene Family

Using five Arabidopsis *AtRFS* gene sequences (*AT1G55740*, *AT3G57520*, *AT4G01970*, *AT5G40390*, and *AT5G20250*) as queries, BLASTp searches (E < 1 × 10^−10^) were conducted against the reference genomes of *B. rapa*, *B. oleracea*, and *B. napus*, resulting in the identification of thirteen *B. napus RFS* genes (*BnaRFS*), six *B. rapa RFS* genes (*BraRFS*), and six *B. oleracea RFS* genes (*BoRFS*). A phylogenetic tree was constructed based on the protein sequences of thirty *RFS* genes from these four species using MEGA11 (Figure 1A). Based on the phylogenetic relationships, the RFS proteins were grouped into three distinct clades: RFS1/2, RFS4/5, and RFS6. Clade I included four *B. napus*, two *A. thaliana*, two *B. rapa*, and two *B. oleracea RFS* genes. Clade II consisted of four *B. napus*, two *A. thaliana*, two *B. rapa*, and two *B. oleracea RFS* genes. Clade III was composed of five *B. napus*, one *A. thaliana*, two *B. rapa*, and two *B. oleracea RFS* genes.

To further investigate the evolutionary characteristics of the *RFS* gene family among *A. thaliana*, *B. napus*, *B. rapa*, and *B. oleracea*, conserved motifs, domains, and gene structures were analyzed and visualized using TBtools-II v2.310. MEME analysis revealed consistent motif compositions and arrangements among the 30 RFS proteins. Except for BnaA10G0175300ZS, which contained only motif 6, 4, 1, 2, and 10, the remaining RFS proteins exhibited similar motif numbers (Figure 1B). Domain analysis using the SMART online tool showed that all members contained the conserved Raffinose_syn domain (Figure 1C). Exon–intron structure analysis demonstrated a conserved pattern within each clade. In Clade I, BnaA09G0529700ZS contained up to 19 exons, and AT3G57520 had 13 exons, while the remaining members had 12 exons. Clade II members contained between 3 and 6 exons. In Clade III, *RFS4* genes consistently had 5 exons, while *RFS5* genes had 4 exons (Figure 1D).

To gain deeper insights into the chromosomal distribution of the RFS gene family, the chromosomal locations of RFS genes in *A. thaliana*, *B. napus*, *B. rapa*, and *B. oleracea* were mapped using TBtools. The 13 *B. napus RFS* genes were distributed across 10 chromosomes (Figure 2A), the 5 *A. thaliana RFS* genes across 4 chromosomes (Figure 2B), the 6 *B. oleracea RFS* genes across 5 chromosomes (Figure 2C), and the 6 *B. rapa RFS* genes across 5 chromosomes (Figure 2D). Notably, *BnaA10G0175100ZS* and *BnaA10G0175300ZS* were located in close proximity on the same chromosome.

### 2.2. Physicochemical Properties of RFS Proteins in B. napus, B. rapa, and B. oleracea

The physicochemical characteristics of RFS protein sequences exhibited considerable variation among different members (Table 1). The amino acid lengths ranged from 425 (BnaA10G0175300ZS) to 1159 residues (BnaA09G0529700ZS), with predicted molecular weights spanning from 46.66 kDa (BnaA10G0175300ZS) to 129.2 kDa (BnaA09G0529700ZS). The theoretical isoelectric points (pI) ranged from 4.94 (BnaC04G0403600ZS) to 6.8 (BnaA09G0529700ZS). The grand average of hydropathicity (GRAVY) values ranged from −0.089 (Bo4g140140) to −0.339 (Bra002286), indicating that most RFS proteins are hydrophilic and neutral in nature. Except for the RFS4 proteins, all others exhibited instability indices below 40, suggesting that these proteins are likely stable in vivo.

### 2.3. Cis-Element Analysis of BnaRFS Gene Promoter Regions

The promoter regions (2000 bp upstream of the start codon) of *BnaRFS* genes were analyzed for cis-acting regulatory elements using the PlantCARE database (https://bioinformatics.psb.ugent.be/webtools/plantcare/html/, accessed on 21 May 2025) and visualized with TBtools. The identified cis-elements were categorized into five groups related to plant hormone responsiveness (ABA, GA, MeJA, SA, and auxin), six types associated with environmental stress responses (light, low temperature, anaerobic, anoxic, circadian, and defense/stress), one group corresponding to transcription factor binding sites (MYB), and four related to tissue- or gene-specific expression (endosperm, meristem, seed-specific, and zein) (Figure 3A). A total of 306 cis-elements were identified across all BnaRFS promoters, including 141 hormone-responsive elements, 119 environmental stress-related elements, and 27 transcription factor binding sites. Among the hormone-responsive elements, those related to MeJA and ABA were the most abundant. For environmental stress-responsive elements, light-responsive and anaerobic-responsive elements were predominant. The MYB binding site was the most frequently identified transcription factor-related cis-element (Figure 3B).

### 2.4. Expression Analysis of BnaRFS Genes in Different Tissues

To further investigate the potential role of *BnaRFS* genes in soluble sugar accumulation, RNA-Seq data were analyzed from various tissues of *B. napus* lines with contrasting soluble sugar contents: a high-soluble-sugar line (No. 51) and a low-soluble-sugar line (No. 106). Samples were collected from leaves at the seedling stage and from leaves, stems, and buds at the bolting stage. The results revealed that *BnaRFS* genes exhibited distinct tissue-specific expression patterns, among which *BnaC02G0100500ZS* showed significant differences across tissues (Figure 4A). To validate the transcriptomic data, the relative expression of *BnaC02G0100500ZS* in lines No. 51 and No. 106 was further examined using qRT-PCR. The qRT-PCR results were largely consistent with the RNA-Seq data, confirming the reliability of the expression profiles (Figure 4B).

### 2.5. Cloning and Subcellular Localization of BnaRFS6

To determine the subcellular localization of the BnaRFS6 protein, the *BnaRFS6* gene was cloned and named based on its homology with *A. thaliana* RFS proteins. A GFP-tagged fusion expression vector was constructed and introduced into Arabidopsis protoplasts via PEG-mediated transformation. Confocal laser scanning microscopy revealed that, compared with the empty vector control, the fluorescence signals of BnaRFS6-GFP co-localized with the mitochondrial marker fluorescence, indicating that BnaRFS6 is localized to the mitochondria (Figure 5).

### 2.6. Overexpression of BnaRFS6 Enhances Soluble Sugar Content in Rapeseed Bolting

To investigate the effect of *BnaRFS6* on sugar accumulation in rapeseed bolting stalks, the *BnaRFS6* gene was cloned into the plant expression vector pC2300 and genetically transformed into *B. napus*. The starch and soluble sugar contents were measured. Compared with the wild type (WT), the soluble sugar content in the *BnaRFS6* overexpression lines (OE-1 and OE-2) was significantly increased (Figure 6A), while starch content was decreased (Figure 6B). Furthermore, the contents of fructose (Figure 6D), glucose (Figure 6E), and raffinose (Figure 6F) were markedly elevated in the overexpression lines. These results indicate that overexpression of *BnaRFS6* increases the total soluble sugar content by promoting the accumulation of fructose, glucose, and raffinose, suggesting that *BnaRFS6* positively regulates soluble sugar accumulation in rapeseed bolting.

## 3. Discussion

Research on the *RFS* gene family in plants remains limited. The first reported *RFS* gene was cloned from pea and was involved in the synthesis of raffinose [16]. Subsequently, *RFS* genes have been cloned and analyzed in cucumber [17] and Arabidopsis [18]. Notably, six *RFS* genes have been identified in both Arabidopsis and rice [19]; however, *AtRFS3* is a pseudogene and was, therefore, excluded from analysis in this study along with its homologs in *B. napus*, *B. rapa*, and *B. oleracea*. Based on the *AtRFS* sequences from the model plant Arabidopsis, this study identified six *BraRFS* genes in *B. rapa*, six *BoRFS* genes in *B. oleracea*, and thirteen *BnaRFS* genes in *B. napus*, all containing the conserved Raffinose_syn domain.

Physicochemical property analysis revealed that all identified *RFS* family members have isoelectric points (pI) below seven and negative grand average hydropathicity values, indicating they are hydrophilic proteins, consistent with observations reported in wheat *RFS* gene family members [20]. Motif analysis showed that all members share a conserved motif arrangement (motif6-motif4-motif1-motif2-motif10), indicating relative conservation of *RFS* genes. Gene structure analysis revealed variation in exon numbers among different subgroups, but most genes within the same subgroup exhibited similar exon–intron organization, in agreement with reports from cotton [21]. This study is the first to confirm mitochondrial localization of BnaRFS6, which differs from maize *ZmRFS3* that primarily localizes to plastids [15]. This phenomenon may be related to the central role of mitochondria in carbon flux reprogramming and energy metabolism. Previous studies have shown that impaired mitochondrial function or altered activity can directly affect the redistribution of sucrose degradation products and the flux of sugars into different metabolic pathways [22].

Analysis of soluble sugar components (including fructose, glucose, sucrose, and raffinose) in *BnaRFS6* overexpressing and wild-type *B. napus* revealed that overexpression of *BnaRFS6* significantly increased the total soluble sugar content in the bolting stems, primarily due to elevated levels of fructose, glucose, and raffinose. This suggests that *BnaRFS6* may regulate carbohydrate metabolism and allocation through multiple mechanisms. These findings are consistent with previous functional studies on the Arabidopsis homolog *AtRFS4*, which has also been shown to be involved in the biosynthesis of raffinose family oligosaccharides (RFOs) [18]. The enhanced expression of *BnaRFS6* may increase the biosynthetic rate of RFOs, leading to the marked accumulation of raffinose. Concurrently, the simultaneous increase in fructose and glucose content alongside raffinose accumulation implies that *BnaRFS6* might influence not only the accumulation of downstream products but also the upstream carbon partitioning or sugar transport processes.

*BnaRFS6* exerts a significant regulatory effect on raffinose metabolism, and *RFS* genes have been shown to participate in plant carbon allocation by modulating raffinose metabolism [23]. Overexpression of *BnaRFS6* led to a marked decrease in starch content and a significant increase in total soluble sugars in the bolting stems of *B. napus*, suggesting that *BnaRFS6* may play a critical role in the dynamic “source–sink” balance. This phenomenon is similar to that observed in potatoes, where overexpression of the *SUT1* gene resulted in reduced starch accumulation and enhanced soluble sugar levels [24], although the underlying regulatory mechanisms may differ. The observed trade-off between soluble sugars and starch implies that *BnaRFS6* may be involved in a key regulatory node controlling carbon skeleton partitioning.

## 4. Materials and Methods

### 4.1. Plant Materials and Growth Conditions

The *B. napus* cultivar used in this study was “Gaoyousuan No. 1”, which was bred by academician Chunyun Guan from Hunan Agricultural University. Plants were grown in a controlled growth chamber with a relative humidity of 60%, a temperature of 21 °C, and a photoperiod of 16 h light/8 h dark.

### 4.2. Bioinformatics Analysis

Genome sequences were obtained from the following databases: *A. thaliana* (https://www.arabidopsis.org/, accessed on 6 January 2025), *B. napus* (http://cbi.hzau.edu.cn/bnapus/index.php, accessed on 6 January 2025), *B. rapa*, and *B. oleracea* (http://brassicadb.cn/#/, accessed on 6 January 2025). The amino acid sequences of Arabidopsis RFS (AtRFS) proteins were used as queries to perform local BLASTP searches against the genomes of *B. napus*, *B. rapa*, and *B. oleracea*. Candidate RFS homologs were identified using an E-value cutoff of <1 × 10^−10^.

Conserved protein domains were identified using the SMART online tool (https://smart.embl.de/, accessed on 8 January 2025). Multiple sequence alignments and phylogenetic trees were generated using MEGA11, and the phylogenetic trees were visually refined using Evolview (https://www.evolgenius.info/evolview/#/, accessed on 8 January 2025). Conserved motifs of RFS proteins from *A. thaliana*, *B. napus*, *B. rapa*, and *B. oleracea* were analyzed using the MEME Suite (https://meme-suite.org/meme/tools/meme, accessed on 9 January 2025). Domain structure analysis was also performed using the NCBI Conserved Domain Database (https://www.ncbi.nlm.nih.gov/Structure/bwrpsb/bwrpsb.cgi, accessed on 9 January 2025).

Promoter cis-acting regulatory elements were analyzed using the PlantCARE database (https://bioinformatics.psb.ugent.be/webtools/plantcare/html/, accessed on 15 January 2025) based on the 2000 bp upstream sequences of the *BnaRFS* genes. Gene structure, chromosomal localization, and cis-element visualization were conducted using TBtools software. The molecular weight (Mw), isoelectric point (pI), instability index, aliphatic index, and grand average of hydropathicity (GRAVY) of the predicted proteins were computed using the Compute pI/Mw tool (https://web.expasy.org/compute_pi/, accessed on 9 January 2025).

### 4.3. Gene Cloning and Subcellular Localization

Based on the reference sequence of *BnaRFS6* (*BnaC02G0100500ZS*), specific primers RFS6-F and RFS6-R were designed (Appendix A). Total RNA was extracted using a Plant Total RNA Extraction Kit (RC411-01, Vazyme Biotech Co., Ltd., Nanjing, China) according to the manufacturer’s protocol. First-strand cDNA synthesis was performed using the TransScript One-Step gDNA Removal and cDNA Synthesis SuperMix kit (AT311, TransGen Biotech, Beijing, China). The full-length *BnaRFS6* coding sequence was amplified from the cDNA template and cloned into the pMDTM19-T vector (6013, Takara Bio Inc., Dalian, China). The recombinant plasmid was then transformed into *Escherichia coli* DH5α via heat shock, and the insert was confirmed by Sanger sequencing. To determine the subcellular localization of the BnaRFS6 protein, the coding sequence was inserted into the pCAMBIA2300-GFP (VT1383, Youbio, Changsha, China) expression vector and transiently expressed in *A. thaliana* mesophyll protoplasts for co-localization analysis. Transient expression in Arabidopsis protoplasts was performed following the method described by Wu et al. [25].

### 4.4. Construction of Overexpression Vector and Genetic Transformation in B. napus

An overexpression vector was constructed based on the pCAMBIA2300-GFP backbone using KpnI and SacI restriction enzyme digestion. Using the sterile seedlings of “Gaoyousuan No. 1” as transformation recipients, surface-sterilized explants were inoculated onto 1/2 MS solid medium for dark cultivation to obtain aseptic seedlings. Hypocotyl segments from these seedlings were excised as explants for Agrobacterium-mediated genetic transformation. After 48 h of co-cultivation, infected (hypocotyl) explants were induced through antibiotic selection, ultimately differentiating into regenerated plants. Genomic DNA was extracted from leaves of regenerated shoots using the CTAB method. Transgenic lines were molecularly identified using overexpression detection primers NPTII-F68 andNPTII-R356 (Appendix A). The confirmed positive transgenic lines were subsequently cultivated to reproductive growth stage. A total of 30% of infected calli generated resistant shoots.

### 4.5. Expression Analysis

Expression data of *BnaRFS* genes were derived from unpublished transcriptome datasets generated in this study. Quantitative real-time PCR (qRT-PCR) was performed using the Taq Pro Universal SYBR qPCR Master Mix (Q712-02, Vazyme Biotech Co., Ltd., Nanjing, China) on a Bio-Rad CFX96 Real-Time PCR System. *BnaActin* was used as the internal reference gene. Relative expression levels were calculated using the 2^−∆∆CT^ method [26].

### 4.6. Sugar Content Determination

Soluble sugar and starch contents were determined using the anthrone colorimetric method [27]. Sucrose, glucose, fructose, and raffinose contents were quantified using high-performance liquid chromatography (HPLC), as described by Filip et al. [28].

### 4.7. Data Analysis

All measurements were performed with three biological replicates. Raw data were organized using WPS Excel (12.1.0.21915). Statistical significance was determined using the least significant difference (LSD) test in SPSS version 26. Differences were considered statistically significant at *p* < 0.05 and highly significant at *p* < 0.01. Graphs were generated using Origin 2021 and TBtools-II v2.310, and figures were further processed using Adobe Illustrator 2023.

## 5. Conclusions

In summary, in the present study, based on the *AtRFS* gene sequence in the model plant *A. thaliana*, bioinformatics methods were used to identify 6 *BraRFS*, 6 *BoRFS*, and 13 *BnaRFS* genes in *B. rapa*, *B. oleracea*, and *B. napus*, respectively, distributed on different chromosomes. Phylogenetic tree analysis showed that the *RFS* gene family was divided into three subgroups. The gene motif results indicate that there are differences in the number of exons between genes in different subgroups, but most *RFS* genes in the same subgroup have the same exon–intron structure. We cloned the *BnaRFS6* gene, and subcellular localization showed that the gene is located on mitochondria. OE-*BnaRFS6* can alter the total soluble sugar content by changing the fructose, glucose, and raffinose content.

## Figures and Tables

**Figure 1 genes-16-01032-f001:**
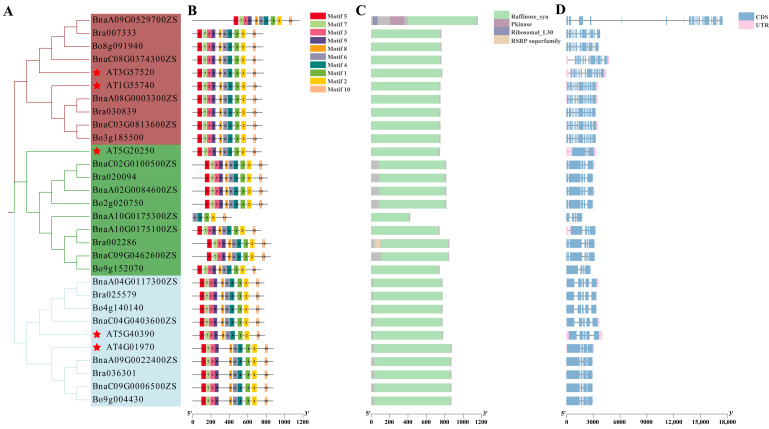
Analysis of neighbor-joining phylogenetic tree (**A**), motif (**B**), domain (**C**), and gene structure (**D**) of *RFS* genes of *A. thaliana*, *B. rapa*, *B. oleracea*, and *B. napus*.

**Figure 2 genes-16-01032-f002:**
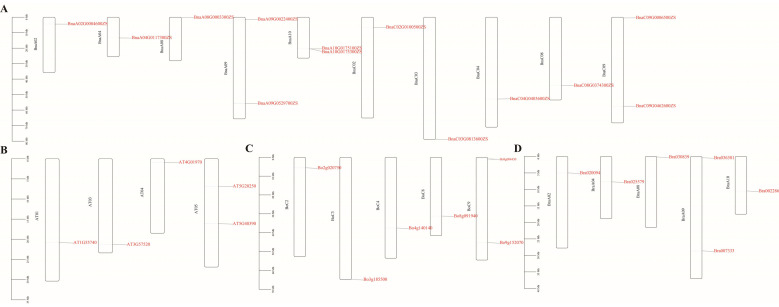
Distribution of *RFS* genes of *B. napus* (**A**), *A. thaliana* (**B**), *B. oleracea* (**C**), and *B. rapa* (**D**).

**Figure 3 genes-16-01032-f003:**
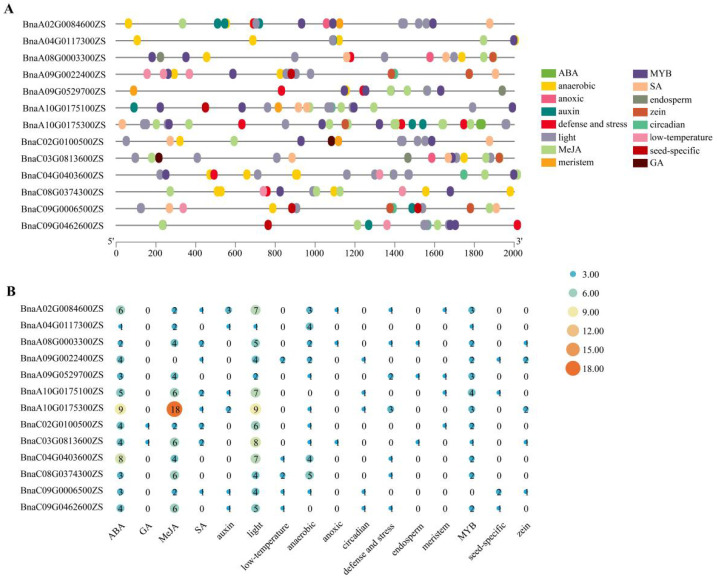
Cis-acting elements in the promoters of *BnaRFS* genes. (**A**) The types of promoter cis-acting elements of *BnaRFS* genes. The different colored boxes represent the different types and positions of cis-acting elements in each *BnaRFS* gene. (**B**) The numbers of promoter cis-acting elements of *BnaRFS* genes.

**Figure 4 genes-16-01032-f004:**
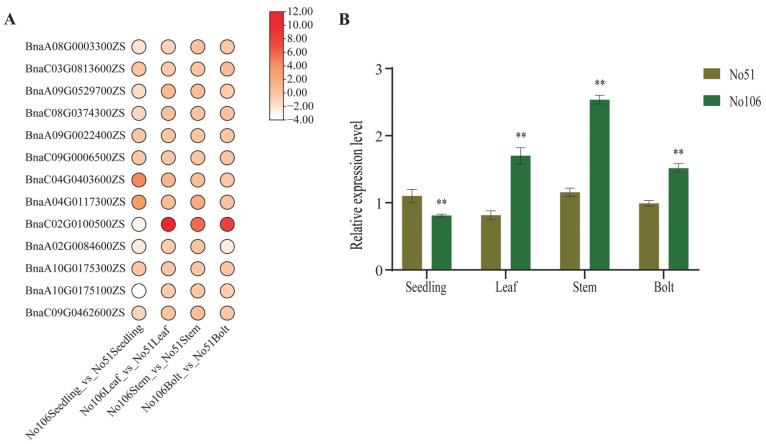
Expression analysis of *BnaRFS* genes. (**A**) Heatmap of *BnaRFS* gene expression in different tissues of *B. napus* based on RNA-Seq data. Seedling: leaves at seedling stage; Leaf: leaves at bolting stage; Bolt: buds at bolting stage; Stem: stems at bolting stage. (**B**) qRT-PCR analysis of *BnaC02G0100500ZS* expression in lines No. 51 and No. 106. The expression level in No51 seedling leaves was set as 1. *BnaActin* was used as the internal reference gene. Data are presented as mean ± standard deviation (SD) from three biological replicates. Statistical significance was assessed using Student’s *t*-test **, *p* < 0.01).

**Figure 5 genes-16-01032-f005:**
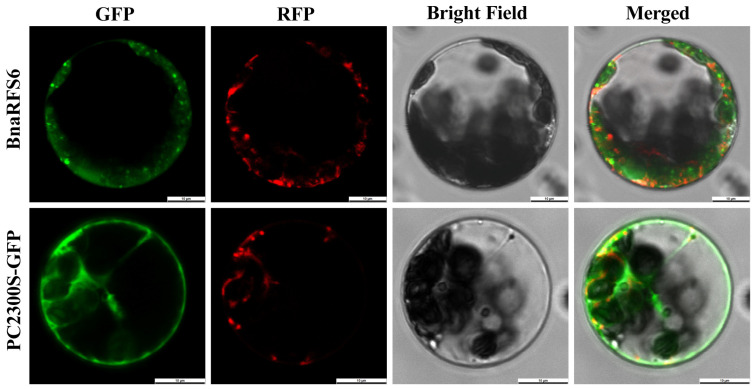
Subcellular localization of BnaRFS6. Green fluorescence represents GFP (green fluorescent protein), and red fluorescence represents RFP (mitochondrial marker). PC2300-GFP serves as the empty vector control.

**Figure 6 genes-16-01032-f006:**
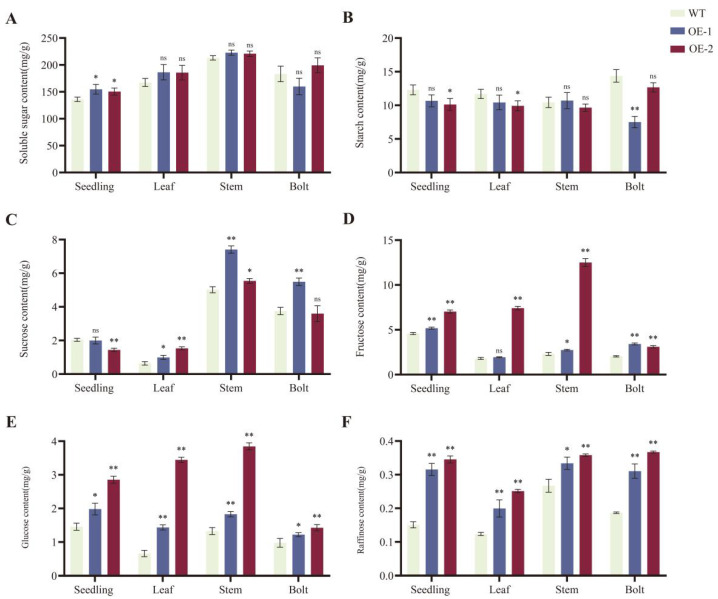
Sugar content analysis in *BnaRFS6* overexpression lines. (**A**) Soluble sugar content. (**B**) Starch content. (**C**) Sucrose content. (**D**) Fructose content. (**E**) Glucose content. (**F**) Raffinose content. WT: wild type; OE-1 and OE-2: *BnaRFS6* overexpression lines. Data are presented as mean ± SD (n = 3). Asterisks indicate significant differences compared to WT, with * and ** representing *p* < 0.05 and *p* < 0.01, respectively; ns indicates no significant difference.

**Table 1 genes-16-01032-t001:** Protein physical and chemical properties of RFS proteins.

Arabidopsis	Homologous Gene	a.a	MW (kDa)	pI	Instability Index	GRAVY
AT1G55740 RFS1	*BnaA08G0003300ZS*	754	83.42	5.94	31.12	−0.194
	*BnaC03G0813600ZS*	754	83.41	6.01	32.22	−0.189
	*Bra030839*	754	83.49	5.93	30.43	−0.199
	*Bo3g185500*	754	83.50	6.04	31.73	−0.204
AT3G57520 RFS2	*BnaA09G0529700ZS*	1159	129.20	6.8	39.04	−0.259
	*BnaC08G0374300ZS*	765	84.03	5.75	36.29	−0.185
	*Bra007333*	764	84.03	5.68	35.05	−0.189
	*Bo8g091940*	765	84.09	5.78	35.69	−0.187
AT4G01970 RFS4	*BnaA09G0022400ZS*	873	97.71	5.12	40.79	−0.236
	*BnaC09G0006500ZS*	873	97.67	5.14	40.09	−0.231
	*Bra036301*	873	97.72	5.14	41.19	−0.239
	*Bo9g004430*	873	97.67	5.14	40.09	−0.231
AT5G40390 RFS5	*BnaC04G0403600ZS*	777	85.87	4.94	33.26	−0.114
	*BnaA04G0117300ZS*	777	85.81	4.96	34.21	−0.12
	*Bra025579*	777	85.81	4.96	34.21	−0.12
	*Bo4g140140*	778	85.92	4.98	33.78	−0.089
AT5G20250 RFS6	*BnaC02G0100500ZS*	816	90.45	6.25	38.26	−0.233
	*BnaA02G0084600ZS*	818	90.50	6.13	39.05	−0.239
	*BnaA10G0175300ZS*	425	46.66	6.06	37.6	−0.159
	*BnaA10G0175100ZS*	744	82.25	5.51	37.28	−0.263
	*BnaC09G0462600ZS*	848	94.13	6.5	42.8	−0.312
	*Bra020094*	816	90.26	6.19	38.05	−0.234
	*Bra002286*	850	94.44	6.23	43.55	−0.339
	*Bo2g020750*	817	90.31	6.08	38.56	−0.253
	*Bo9g152070*	745	82.38	5.8	39.35	−0.252

## Data Availability

All the data included in this study are available upon request by contacting the corresponding author.

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
