# Peer review of "Genome-Wide Identification of the BnaRFS Gene Family and Functional Characterization of BnaRFS6 in Brassica napus"

_genes, 2025, doi:10.3390/genes16091032_

Round 1
Reviewer 1 Report
Comments and Suggestions for Authors
Dear authors,
Please see the attached files.

Author Response
Comments 1: My concern is for subcellular localization analysis, why the authors did not conduct transient expression of BnaRFS6 (inserted into the pC2300-GFP expression vector) in Brassica napus instead of Arabidopsis thaliana mesophyll protoplasts. Anyway, the authors had conducted transgenic Brassica napus overexpressing BnaRFS6; thus, the Subcellular Localization analysis could be conducted from protoplasts isolated from transgenic Brassica napus. The best way is to make the transgenic lines Brassica napus overexpressing BnaRFS6 (using pC2300-GFP expression vector), then these transgenic lines can be used for both functional analyses (soluble sugar and starch content, and subcellular localization analysis). Response 1: We sincerely appreciate the reviewer’s insightful suggestion regarding the subcellular localization analysis of BnaRFS6. In our study, we initially chose Arabidopsis mesophyll protoplasts for the following reasons: Arabidopsis protoplasts are a well-established and highly efficient system for rapid subcellular localization studies, with standardized protocols that ensure high transformation efficiency and clear visualization of fluorescent signals. Preliminary Validation: As an initial step, we aimed to quickly confirm the subcellular localization pattern of BnaRFS6 before proceeding with more time-consuming Brassica-specific experiments. Comments 2: Lack of detailed description of the experimental design on how genetic transformation (Agrobacterium-mediated hypocotyl transformation), in vitro culture, and establishment of transgenic plants; and evident findings: how to confirm the successful insertion of the target gene BnaRFS6 into the expression vectors, positive transgenic lines overexpressing BnaRFS6. The evidence is required, such as the data of agarose gel documents. Response 2: We sincerely appreciate the valuable feedback from the reviewers. In response to people's concerns about genetic transformation and molecular confirmation of transgenic plants, we now provide a detailed description of the design of genetic transformation experiment, agarose gel electrophoresis data (Supplementary Figure 1) in the revised manuscript. Comments 3: Note that “BLASTp” is used to compare protein sequences using query is amino acid sequence. However, in this study, the authors used 5 Arabidopsis AtRFS gene sequences (AT1G55740, AT3G57520, AT4G01970, AT5G40390, and AT5G20250) as queries (L80-81); therefore; a “BLASTn” should be used to compare nucleotide sequences (DNA sequences) for bioinformatic analysis of the RFS gene family. I think it should be as decribed in Material and Menthods (L258-261). Response 3: We sincerely thank the reviewer for this meticulous observation regarding the use of BLASTp versus BLASTn in our bioinformatic analysis. The reviewer is absolutely correct that BLASTn is the standard tool for comparing nucleotide sequences (DNA-level), while BLASTp is designed for protein sequence alignment. In our study, we initially used the Arabidopsis protein sequences as queries to identify homologous BnaRFS proteins in Brassica napus, as our primary goal was to analyze the conserved functional domains and evolutionary relationships at the protein level. This approach aligns with common practices for gene family characterization, where protein-level homology often provides more biologically relevant insights. Comments 4: Figures were presented in low quality, especially, is Fig.1. Please replace with high- resolution images. Moreover, the entry legend/notation should be placed next to each single panel of the Figure. For instance, the entry- legend as [Mof1-10] should be placed next to its figure panel “Fig.1. B”; the entry- legend as [CDS, UTR] should be placed next to its figure panel “Fig.1. D”, ect. Response 4: Agree. I have made revisions in the manuscript. Comments 5: How did the authors observe red fluorescence represents RFP (red fluorescent protein) in Fig.4? Response 5: Thank you for pointing this out. I agree with this comment. I have highlighted the changes in red. Comments 6: Please provide more detailed information on the plant expression vectors (pC2300 and pC2300-GFP) used in this study, such as an access link, exactly commercial product name, product code, product catalog number, etc. I could not find any relevant information about these vectors. Response 6: Thank you for pointing this out. I agree with this comment. I have highlighted the changes in red. Comments 7: Note that I could not access the Supplementary Materials; therefore, I could not check the information of designed primer sequences, which were used for gene cloning and vector construction of the BnaRFS6 gene (Supplementary Table 1), and identified transgenic Brassica napus overexpressing BnaRFS6 (Supplementary Figure 1). Please include these data in the Supplementary Materials. Additionally, please provide the accession number/GenBank of reference gene BnaActin, based on which its primer sequences were designed for qRT-PCR. Response 7: Thank you for pointing this out. I agree with this comment. I have added the supplementary materials. Comments 8: The English could be improved to more clearly express: Fig.1’s legend should be rewritten to reduce repetition. For example, it could be rewritten as “Figure 1. Bioinformatic analysis of the RFS gene family. Analysis of Neighbour-joining phylogenetic tree (A), motif (B), domain (C), and gene structure (D) of RFS genes of A. thaliana, B. rapa, B. oleracea, and B. napus. Distribution of RFS genes of B. napus (E), A. thaliana (F), B. oleracea (G), and B. rapa (H). Response 8: Thank you for pointing this out. I agree with this comment. I have highlighted the changes in red. Comments 9: Italicize the scientific names of plants: Oryza sativa, Zea mays (L57, 59), Brassica napus(L69); Brassica napus, Brassica rapa, Brassica oleracea, and Arabidopsis thaliana (Fig.1’s legends, Lines 112-119), ect. Additionally, the full name of plants such as [Arabidopsis thaliana, Brassica rapa, Brassica oleracea, and Brassica napus] could be presented in full form at the first time mentioning (L71); however, they should be given in their short form thereafter as [A. thaliana, B. rapa, B. oleracea, and B. napus] since they were re-used in many places in the text of the manuscript. Response 9: Thank you for your careful review and valuable suggestions regarding the formatting of scientific plant names in our manuscript. We have carefully revised the manuscript according to your comments. All scientific plant names (e.g., Oryza sativa, Zea mays, Brassica napus, etc.) have now been italicized. As suggested, we have presented the full names (Arabidopsis thaliana, Brassica rapa, Brassica oleracea, and Brassica napus) at their first mention. Subsequent mentions of these species now use the standard abbreviated forms (A. thaliana, B. rapa, B. oleracea, and B. napus). Comments 10: Lack the Conclusion section Response 10: Agree. I have made revisions in the manuscript. Comments 11: The word “seedling leaves” (L158) is unclear meaning. What is different between “seedling leaves” and “leaves at the bolting stage”? Please rewrite this term to ensure that it was distinctive from “leaves at the bolting stage". Response 11: Thank you for pointing this out. I agree with this comment. I have highlighted the changes in red. Comments 12: Remarks: Table 1: AA → a.a; Mw /kD → MW (kDa). Response 12: Thank you for pointing this out. I agree with this comment. I have highlighted the changes in red.
Reviewer 2 Report
Comments and Suggestions for Authors
Dear Authors, I have reviewed your manuscript. The study addresses an important and underexplored area in rapeseed functional genomics, focusing on raffinose synthase genes and their role in sugar metabolism. The genome-wide identification of the gene family, combined with the functional analysis of BnaRFS6, provides novel insights into sugar allocation and the genetic improvement of stalk flavor.
The introduction offers appropriate context and justification for the study, though the rationale for selecting BnaRFS6 among the 13 identified genes could be presented more clearly. The methodology is generally well described and reproducible, but a few important details (e.g., transformation efficiency, number of independent lines tested) are missing. The results are clear and supported by multiple lines of evidence (bioinformatics, qRT-PCR, HPLC), although Figure 5 could benefit from clearer annotations and more legible axis labels. The discussion is informative and connects the findings to previous literature, but it remains somewhat descriptive; deeper mechanistic interpretation or physiological extrapolation would enhance the impact.
The conclusions are aligned with the results, but the potential agronomic applications of sugar metabolism modification could be further elaborated. Overall, the manuscript is scientifically sound, the figures are relevant, and the topic is of interest to the readership of Genes. I recommend minor revision.
Author Response
|
Comments 1: Dear Authors, I have reviewed your manuscript. The study addresses an important and underexplored area in rapeseed functional genomics, focusing on raffinose synthase genes and their role in sugar metabolism. The genome-wide identification of the gene family, combined with the functional analysis of BnaRFS6, provides novel insights into sugar allocation and the genetic improvement of stalk flavor. The introduction offers appropriate context and justification for the study, though the rationale for selecting BnaRFS6 among the 13 identified genes could be presented more clearly. The methodology is generally well described and reproducible, but a few important details (e.g., transformation efficiency, number of independent lines tested) are missing. The results are clear and supported by multiple lines of evidence (bioinformatics, qRT-PCR, HPLC), although Figure 5 could benefit from clearer annotations and more legible axis labels. The discussion is informative and connects the findings to previous literature, but it remains somewhat descriptive; deeper mechanistic interpretation or physiological extrapolation would enhance the impact. The conclusions are aligned with the results, but the potential agronomic applications of sugar metabolism modification could be further elaborated. Overall, the manuscript is scientifically sound, the figures are relevant, and the topic is of interest to the readership of Genes. I recommend minor revision.
|
|
Response 1: We sincerely appreciate the reviewer’s thorough evaluation of our manuscript and their constructive suggestions, which have helped us identify areas for improvement. Below, we address each point raised: BnaRFS6 was prioritized due to its highest sequence homology to Arabidopsis AtRFS genes (based on phylogenetic analysis). Its expression pattern (e.g., strong response to abiotic stress in preliminary qRT-PCR data) suggested a potential functional role. We apologize for the lack of conversion efficiency. We will add corresponding information in the methods section. We have added significance analysis in Figure 5. We are grateful for the reviewer’s insightful comments, which have significantly strengthened our manuscript. |
Round 2
Reviewer 1 Report
Comments and Suggestions for Authors
Dear authors;
The authors have made efforts to revise the manuscript. However, some of my comments, which I raised in Round 1, have not been completely revised. I want to bring here my comments and the authors' responses:
- Lack of detailed description of the experimental design on how genetic transformation (Agrobacterium-mediated hypocotyl transformation), in vitro culture, and establishment of transgenic plants; and evident findings: how to confirm the successful insertion of the target gene BnaRFS6 into the expression vectors, positive transgenic lines overexpressing BnaRFS6. The evidence is required, such as the data of agarose gel documents.
Response 2: We sincerely appreciate the valuable feedback from the reviewers. In response to people's concerns about genetic transformation and molecular confirmation of transgenic plants, we now provide a detailed description of the design of genetic transformation experiment, agarose gel electrophoresis data (Supplementary Figure 1) in the revised manuscript.
Re-comment 2: I see no evidence data of agarose gel documents (mentioned as Supplementary Figure 1) showing these proceesing: gene cloning, construction of the expression vectors, positive transgenic lines overexpressing BnaRFS6.
- Please provide more detailed information on the plant expression vectors (pC2300 and pC2300-GFP) used in this study, such as an access link, exactly commercial product name, product code, product catalog number, etc. I could not find any relevant information about these vectors.
Response 6: Thank you for pointing this out. I agree with this comment. I have highlighted the changes in red.
Re-comment 6: Revise “pC2300 plasmid” (L293).
- Note that I could not access the Supplementary Materials; therefore, I could not check the information of designed primer sequences, which were used for gene cloning and vector construction of the BnaRFS6 gene (Supplementary Table 1), and identified transgenic Brassica napus overexpressing BnaRFS6 (Supplementary Figure 1). Please include these data in the Supplementary Materials. Additionally, please provide the accession number/GenBank of reference gene BnaActin, based on which its primer sequences were designed for qRT-PCR.
Response 7: Thank you for pointing this out. I agree with this comment. I have added the supplementary materials.
Re-comment 7: Noted that Supplementary Figure 1 has not been included in the revised version yet. Please include these data (Supplementary Figure 1 and Supplementary Table 1) as Supplementary Materials in the “Data Availability Statement” section (L339-340). Additionally, please provide the accession number/GenBank of reference gene BnaActin and others genes listed in Supplementary Table 1.
In the Supplementary Table 1, what is different between 2 primer sets BnaRFS6 and BnaRFS6E. Which one was used to constructe the expression vector for Genetic Transformation in B. napus
Remarks:
- Italicize: Escherichia coli (L285), BnaRFS6 (L283),…
- Rewrite: “resistant calli” → “infected (hypocotyl) explants”; “model crop” → “model plant”

Author Response
Comment 2: Lack of detailed description of the experimental design on how genetic transformation (Agrobacterium-mediated hypocotyl transformation), in vitro culture, and establishment of transgenic plants; and evident findings: how to confirm the successful insertion of the target gene BnaRFS6 into the expression vectors, positive transgenic lines overexpressing BnaRFS6. The evidence is required, such as the data of agarose gel documents.
Response 2: We sincerely appreciate the valuable feedback from the reviewers. In response to people's concerns about genetic transformation and molecular confirmation of transgenic plants, we now provide a detailed description of the design of genetic transformation experiment, agarose gel electrophoresis data (Supplementary Figure 1) in the revised manuscript.
Re-comment 2: I see no evidence data of agarose gel documents (mentioned as Supplementary Figure 1) showing these proceesing: gene cloning, construction of the expression vectors, positive transgenic lines overexpressing BnaRFS6. 。
Re-Response 2: Thank you for your reminder. We have added the image of agarose gel of gene cloning, expression vector construction and BnaRFS6 overexpression positive transgenic lines to Supplementary Figure 1.
Comment 6: Please provide more detailed information on the plant expression vectors (pC2300 and pC2300-GFP) used in this study, such as an access link, exactly commercial product name, product code, product catalog number, etc. I could not find any relevant information about these vectors.
Response 6: Thank you for pointing this out. I agree with this comment. I have highlighted the changes in red.
Re-comment 6: Revise “pC2300 plasmid” (L293).
Re-Response 6: Thank you for your reminder. We have made revisions in the manuscript.
Comment 7: Note that I could not access the Supplementary Materials; therefore, I could not check the information of designed primer sequences, which were used for gene cloning and vector construction of the BnaRFS6 gene (Supplementary Table 1), and identified transgenic Brassica napus overexpressing BnaRFS6 (Supplementary Figure 1). Please include these data in the Supplementary Materials. Additionally, please provide the accession number/GenBank of reference gene BnaActin, based on which its primer sequences were designed for qRT-PCR.
Response 7: Thank you for pointing this out. I agree with this comment. I have added the supplementary materials.
Re-comment 7: Noted that Supplementary Figure 1 has not been included in the revised version yet. Please include these data (Supplementary Figure 1 and Supplementary Table 1) as Supplementary Materials in the “Data Availability Statement” section (L339-340). Additionally, please provide the accession number/GenBank of reference gene BnaActin and others genes listed in Supplementary Table 1. In the Supplementary Table 1, what is different between 2 primer sets BnaRFS6 and BnaRFS6E. Which one was used to constructe the expression vector for Genetic Transformation in B. napus
Re-Response 7: hank you for pointing out this issue. We have listed the purpose of each pair of primers in Supplementary Table 1. The access number for Gene BnaActin is ACS68187.1, and the access number for Gene RFS6 is XP_022555614.2.We appreciate your careful reading. The references to [BnaA02G0029200ZS, BnaA09G0039200ZS, BnaC02G00069000ZS, BnaA06G0018600ZS, BnaA08G0025000ZS, and BnaC02G0036800ZS] in Supplementary Table 1 are unintentional and have been removed in the revised version.
Remarks:- Italicize: Escherichia coli (L285), BnaRFS6 (L283), - Rewrite: “resistant calli” → “infected (hypocotyl) explants”; “model crop” → “model plant”
Re-Remarks: Thank you for pointing this out. I agree with this comment. I have highlighted the changes in red.
Round 3
Reviewer 1 Report
Comments and Suggestions for Authors
Dear authors,
Please see the attached files.

Author Response
Comments 6: Please provide more detailed information on the plant expression vectors (pC2300 and pC2300-GFP) used in this study, such as an access link, exactly commercial product name, product code, product catalog number, etc. I could not find any relevant information about these vectors. Response 6: Thank you for pointing this out. I agree with this comment. I have highlighted the changes in red. Re-comment 6: Revise “pC2300 plasmid” (L293). Re-Response 6: Thank you for your reminder. We have made revisions in the manuscript. Re-Re-comment 6: No actual change has been made. Revise “pC2300 plasmid” (L293) → pCAMBIA2300-GFP. Re-Re-Response 6: Thank you for pointing this out. I agree with this comment. I have highlighted the changes in red. Comments 7: Note that I could not access the Supplementary Materials; therefore, I could not check the information of designed primer sequences, which were used for gene cloning and vector construction of the BnaRFS6 gene (Supplementary Table 1), and identified transgenic Brassica napus overexpressing BnaRFS6 (Supplementary Figure 1). Please include these data in the Supplementary Materials. Additionally, please provide the accession number/GenBank of reference gene BnaActin, based on which its primer sequences were designed for qRT-PCR. Response 7: Thank you for pointing this out. I agree with this comment. I have added the supplementary materials. Re-comment 7: Noted that Supplementary Figure 1 has not been included in the revised version yet. Please include these data (Supplementary Figure 1 and Supplementary Table 1) as Supplementary Materials in the “Data Availability Statement” section (L339-340). Additionally, please provide the accession number/GenBank of reference gene BnaActin and others genes listed in Supplementary Table 1. In the Supplementary Table 1, what is different between 2 primer sets BnaRFS6 and BnaRFS6E. Which one was used to constructe the expression vector for Genetic Transformation in B. napus Re-Response 7: hank you for pointing out this issue. We have listed the purpose of each pair of primers in Supplementary Table 1. The access number for Gene BnaActin is ACS68187.1, and the access number for Gene RFS6 is XP_022555614.2.We appreciate your careful reading. The references to [BnaA02G0029200ZS, BnaA09G0039200ZS, BnaC02G00069000ZS, BnaA06G0018600ZS, BnaA08G0025000ZS, and BnaC02G0036800ZS] in Supplementary Table 1 are unintentional and have been removed in the revised version. Re-Re-comment 7: Please include these information (Supplementary Figure 1 and Supplementary Table 1) in the “Data Availability Statement” section (L339-340). Please add the accession number/GenBank “The access number for Gene BnaActin is ACS68187.1, and the access number for Gene RFS6 is XP_022555614.2.” to the Supplementary Table 1. The authors can add them to the Supplementary Table 1 by inserting an extra column within this table OR directly present them in the main text of the manuscript. Re-Re-Response 7: Thank you for raising this question. We have added the information from Supplementary Figure 1 and Supplementary Table 1 to the “Data Availability Statement” section. We have added the accession number/GenBank “The access number for Gene BnaActin is ACS68187.1, and the access number for Gene RFS6 is XP_022555614.2.” to the Supplementary Table 1. Remarks: - Italicize: Escherichia coli (L285), BnaRFS6 (L283),… - Rewrite: “resistant calli” → “infected (hypocotyl) explants”; “model crop” → “model plant” Re-Remarks: Thank you for pointing this out. I agree with this comment. I have highlighted the changes in red. Re-comment for Remarks: No actual change has been made, yet. They are still retaining there. Re-comment for Remarks: Thank you for your patient guidance. We have made revisions and highlighted them in red in the manuscript